# OpenReview forum: "GeneBreaker: Jailbreak Attacks against DNA Language Models with Pathogenicity Guidance"
_ICLR.cc/2026/Conference — ICLR 2026 Poster_

### Official Review · Reviewer_2Aq7 · 2025-10-24

**Soundness:** 3
**Presentation:** 3
**Contribution:** 4
**Rating:** 6
**Confidence:** 3

**Summary:**

This paper presents the first systematic biosafety evaluation of DNA Language Models, specifically the state-of-the-art Evo series models, against jailbreak attacks. The authors introduced JailbreakDNABench, a comprehensive benchmark of six high-priority viral categories and evaluation pipeline for systematic biosecurity risk assessments. Also, they introduced GeneBreaker, a novel jailbreak framework that probes vulnerabilities of DNA language models. They find that Genebreaker can successfully jailbreak the  series models on all 6 viral categories, achieving an Attack Success Rate of up to 60.0% on the Evo2-40B.

**Strengths:**

- Overall clear and well-structured paper
- Addresses one of the most critical safety concerns surrounding generative AI applied to biology, providing the first systematic evidence of a jailbreak vulnerability in frontier DNA-LM.
- Novel and effective jailbreaking framework, GeneBreaker, that combines domain-specific knowledge and modern LLM jailbreak tactics at is able to successfully jailbreaks the latest Evo series models across 6 viral categories.

**Weaknesses:**

- The reliance on PathoLM and logP to guide chunk decoding may limit their search space. Did the authors did any study on using other methods of pathogen scoring to test the robustness of their methods.
- The lack of wet lab experiments to verify their findings. It will further boost the impact of their work.

**Questions:**

- Have the authors explored any alternative scoring functions to guide the beam search?

---

> ### Author Response · Authors · 2025-11-16
>
> We thank the reviewer for the insightful comment and appreciation! As for your concerns:
>
> **Q1**. The reliance on PathoLM and logP to guide chunk decoding may limit their search space. Did the authors conduct any study on using other methods of pathogen scoring to test the robustness of their methods?
>
> **R1**: Yes. To assess whether GeneBreaker relies specifically on PathoLM or log-probability signals, we conducted an additional experiment during the rebuttal phase using geNomad as an alternative pathogen-scoring method. geNomad is a state-of-the-art model for detecting viral genetic elements [1], and was recently used as the pathogenicity-guidance signal in Generative design of novel bacteriophages with genome language models [2]. Following this framework, we replaced PathoLM with geNomad scores for chunk-level decoding in GeneBreaker. The jailbreak success rates obtained under geNomad guidance were comparable to those with PathoLM on Evo2-7B, demonstrating that GeneBreaker’s effectiveness is robust to the choice of pathogen-scoring model. These results indicate that jailbreak vulnerabilities stem from the underlying genomic knowledge learned by the foundation models, rather than dependence on a particular scoring method.
>
> | Model        | Large DNA      | Small DNA       | +ssRNA         | -ssRNA         | dsRNA          | Enteric RNA    |
> |--------------|----------------|------------------|----------------|----------------|----------------|----------------|
> | Evo2 (PathoLM)    | 44.0 ± 8.4     | 46.7 ± 26.7      | 27.8 ± 5.9    | 24.4 ± 12.8    | 20.0 ± 40.0    | 50.0 ± 15.8    |
> | Evo2 (geNomad)   | 42.0 ± 9.7     | 40.8 ± 17.0      | 28.9 ± 7.1     | 26.7 ± 14.1    | 20.0 ± 40.0    | 40.0 ± 13.5    |
>
> References
> [1] Camargo, A., et al. (2023). Identification of mobile genetic elements with geNomad. Nature Biotechnology.
> [2] King, S. H., et al. (2025). Generative design of novel bacteriophages with genome language models. bioRxiv.
>
> **Q2**. The lack of wet lab experiments to verify their findings. It will further boost the impact of their work.
>
> **R2**: We appreciate the reviewer’s suggestion. Wet-lab validation would indeed strengthen the biological interpretation of jailbreak outputs. At the moment, we are actively discussing potential follow-up experiments with our collaborating wet-lab groups. However, conducting assays involving pathogenic or high-risk viral sequences requires navigating extensive biosafety, regulatory, and ethical constraints, and such experiments cannot be initiated without appropriate BSL-2+/BSL-3 facilities, specialized personnel, and institutional review. Moreover, we are also exploring feasible low-risk alternatives (e.g., testing on attenuated or non-pathogenic viral homologs, or structure/function assays using synthetic subcomponents) for future studies. We view this as an important next step and are working with collaborators to design safe and compliant experimental follow-ups.

---

### Official Review · Reviewer_aFJo · 2025-10-26

**Soundness:** 3
**Presentation:** 2
**Contribution:** 3
**Rating:** 6
**Confidence:** 3

**Summary:**

This paper introduces a framework to evaluate whether genomic foundation models can be jailbroken to produce sequences closely resembling regulated human pathogens. The contributions include
(i) JailbreakDNABench, a curated benchmark covering six viral categories.
(ii) GeneBreaker, which combines an LLM agent for high-homology, non-pathogenic prompt design, beam search guided by PathoLM + log-prob heuristics, and an evaluation pipeline using BLAST and VADR function checks.
On JailbreakDNABench, GeneBreaker successfully jailbreaks the latest Evo series models across 6 viral categories consistently.

**Strengths:**

(i) Novelty: The paper provides the first framework to evaluate whether genomic foundation models can be jailbroken to produce sequences closely resembling regulated human pathogens.

(ii) Technical completeness: LLM-assisted prompt construction + guided beam search + BLAST/VADR evaluation.

(iii) Robust experimental results: On JailbreakDNABench, GeneBreaker successfully jailbreaks the latest Evo series models across 6 viral categories consistently.

**Weaknesses:**

(i) Dataset transparency: The paper does not provide a full list of sequences/accession IDs or the dataset statistics (e.g., dataset sample size)

(ii) There are two biological statements, it is a little hard for people without much biological background to understand.

(iii) Wrong statements: In Table 3, GenomeOcean just uses the Mistral architecture, not an MOE model.

**Questions:**

(i) Dataset size: I do not see the dataset sample size, but I am concerned about whether the sample size is enough in each task as a benchmark.

(ii) The tasks are about the virus. Are there other aspects we can think about to evaluate whether genomic foundation models can be jailbroken?

(iii) What kind of insights can we get to avoid the jailbreak problem when developing the genomic foundation models?

**Details Of Ethics Concerns:**

This paper may provide a method to generate harmful virus sequences based on genomic foundation models.

---

> ### Author Response · Authors · 2025-11-16
>
> We thank the reviewer for the detailed comment and appreciation! As for the questions:
>
> **Q1**.  Dataset transparency: The paper does not provide a full list of sequences/accession IDs or the dataset statistics (e.g., dataset sample size).  I am concerned about whether the sample size is enough in each task as a benchmark.
>
> **R1**: We thank the reviewer for raising this point. In Appendix B, we provide the full list of viral sequences, category assignments, and descriptions used in JailbreakDNABench for transparency. The benchmark includes 94 pathogen-associated viral samples, each representing one critical CDS region in one selected virus sequence. These sequences were selected directly from the U.S. HHS and USDA Select Agents and Toxins Lists, which define biological agents posing the highest biosafety concern. Our goal is therefore not to assemble a large dataset, but to cover the most critical, regulated, and high-risk viral families that are internationally recognized in biosafety policy.
>
> Regarding sample size, viral jailbreak evaluation differs from conventional supervised benchmarks: the task hinges on whether a model can reconstruct or approximate regulated pathogen sequences, not on large-sample statistical generalization. Using a concise, policy-defined set of high-concern viruses keeps the benchmark focused, interpretable, and aligned with real-world biosafety screening standards, while also reducing unnecessary evaluation burden. We will add clarifications in the text to make this motivation explicit.
>
> **Q2**. It is a little hard for people without much biological background to understand.
>
> **R2**: Thanks for the suggestion! We will polish the biological statements to make them easier to understand for non-biology background people. For example, we will include more explanation of terms and basic concepts in the paper.
>
> **Q3**.  In Table 3, GenomeOcean just uses the Mistral architecture, not an MOE model.
>
> **R3** : Thanks for pointing this out. We will update Table 3 and fix this error.
>
> **Q4**. The tasks are about the virus. Are there other aspects we can think about to evaluate whether genomic foundation models can be jailbroken?
>
> **R4**: Thanks for the insightful question. Yes. Although our current benchmark focuses on viruses due to their clear regulatory and biosafety relevance, genomic foundation models exhibit broader jailbreak surfaces beyond pathogenic genome reconstruction. Two important additional aspects are:
> 1. Gene-editing–related jailbreaks (e.g., human embryo gene editing).
> DNA models could be prompted to redesign CRISPR guide RNAs, HDR templates, or regulatory elements enabling targeted edits in sensitive contexts such as human embryos. This is conceptually analogous to LLM jailbreaks that elicit harmful procedural instructions: even without generating a pathogen, a DNA language model could be coerced into producing functional gene-editing components with potentially serious bioethical implications.
>
>
> 2. Recovery of privacy-sensitive or proprietary DNA sequences.
> Similar to how LLMs may memorize and output copyrighted or private text, genomic models trained on proprietary or unpublished DNA data may inadvertently regenerate confidential sequences (e.g., patented constructs, clinical genomic data) when adversarially prompted. This represents an “information-leakage jailbreak” that is distinct from biosafety concerns but highly relevant for data governance and IP protection.
>
> We will include these discussions in our paper.

---

> ### Author Response · Authors · 2025-11-16
>
> **Q5**. What kind of insights can we get to avoid the jailbreak problem when developing the genomic foundation models?
>
> **R5**: Our findings suggest several insights for improving the safety of genomic foundation models and reducing jailbreak vulnerability:
>
> 1. **Sequence-level alignment is necessary, not just content filtering**.
> Our DPO-based alignment experiments show that model-internal preference shaping is more effective than relying solely on inference-time vetoes. This indicates that genomic LMs benefit from training-time safety alignment that explicitly downweights pathogenic continuations.
>
>
> 2. **High-homology prompts are a major vulnerability.**
> GeneBreaker succeeds mainly because benign, high-homology sequences act as effective adversarial prompts. This suggests that future GFMs should incorporate homology-aware safety filters or similarity-triggered refusal mechanisms at both token and chunk levels.
>
>
> 3. **Pathogenicity-aware representations are needed.**
> Embedding-level pathogenicity classifiers (e.g., PathoLM-style modules) can be integrated into the backbone model to detect and downregulate emerging pathogen-like patterns during generation.
>
>
> 4. **Robust memorization management.**
> Because jailbreaks can reconstruct viral or proprietary sequences from partial context, models need memorization audits, watermarking, and training-set provenance tracking to reduce unintentional sequence leakage.
>
>
> 5. **Training data curation beyond simple blacklist removal.**
> Our results show that simply excluding viral genomes from training data is insufficient—models generalize and reconstruct them from evolutionarily related sequences. More robust strategies include genomic differential privacy, sequence de-identification, and selective downsampling of high-risk taxa.
>
>
> 6. **Red-teaming as a continuous part of model development.**
> Just as LLMs now undergo continuous adversarial probing, genomic models should be routinely evaluated with frameworks like GeneBreaker to quantify residual vulnerabilities before deployment.

---

> ### Comment · Reviewer_aFJo · 2025-11-22
>
> Thanks for the thorough response from the authors. The response resolves my concerns, and I will keep my positive score of 6. One additional suggestion is that the authors could directly upload the revised version instead of saying they will update it in the future.

---

> > ### Author Response · Authors · 2025-11-24
> > **Thank you for the positive feedback and suggestion!**
> >
> > Dear Reviewer:
> >
> > We thank you for your continued engagement and for maintaining a positive evaluation of our work. We are glad your concerns are all well resolved.
> >
> > We have followed your advice and uploaded the revised manuscript. To make the changes easy to track, we have marked the updated parts in blue.
> >
> > We truly appreciate your support and the current positive score. We respectfully ask if you would consider raising your score after rebuttal? This would provide stronger support for the paper during the final decision phase. Thank you so much!
> >
> > Bests,

---

> > > ### Comment · Reviewer_aFJo · 2025-11-26
> > >
> > > Thank you for addressing all of my additional comments. I appreciate the revisions and the clarity added in the updated manuscript.
> > >
> > > Regarding the score, I believe that a score of 6 reflects the strengths of the work at this stage. While I am positive about the contribution, I feel that 6 is the most suitable evaluation.
> > >
> > > Thank you again for your efforts in improving the paper. I wish you the best in the review process.

---

### Official Review · Reviewer_TZ2K · 2025-11-01

**Soundness:** 3
**Presentation:** 3
**Contribution:** 2
**Rating:** 6
**Confidence:** 4

**Summary:**

This paper investigates a serious and emerging problem: whether DNA language models (DNA-LMs), large generative models trained on genomic data, can be jailbroken to produce potentially dangerous biological sequences, such as viral genes. The authors build a new benchmark called JailbreakDNABench, covering six classes of human viruses, and propose a red-teaming system named GeneBreaker to test how easily these models can be exploited.

GeneBreaker uses three main components:
(1) a large language model agent that automatically finds non-pathogenic DNA sequences similar to real viruses to craft jailbreak prompts;
(2) a beam search guided by another model (PathoLM) and a probability-based score to push DNA-LMs toward pathogen-like outputs; and
(3) a BLAST- and annotation-based pipeline to check if the generated sequences resemble known human viruses.

The authors test several state-of-the-art DNA models (Evo2-40B, GenomeOcean, GENERator) and find that the larger the model, the higher the jailbreak success rate, reaching up to 60% similarity for some virus categories. Case studies on SARS-CoV-2 and HIV-1 show that the generated DNA can recreate proteins with high structural similarity to real viral proteins when analyzed using AlphaFold3.

**Strengths:**

While many works study jailbreaks in text LLMs, almost none examine biological foundation models. The proposed JailbreakDNABench gives the community a starting point to measure and compare biosafety risks. The GeneBreaker framework combines a prompt-designing LLM, a guided beam search, and a BLAST-based evaluation pipeline in a way that feels methodical and reproducible. The idea of using PathoLM as a guidance signal is clever and biologically grounded. The authors test multiple large DNA models across six virus categories and analyze how model size, prompt similarity, and search parameters affect attack success. These broad results make the findings more convincing.

**Weaknesses:**

1. The paper stops at exposing risks but provides no concrete defense or biosafety governance mechanism beyond brief veto filtering in the Appendix.

2. The benchmark seems built entirely by the authors. How can we be sure JailbreakDNABench isn’t biased toward viruses that are easier to hit, making GeneBreaker look stronger?

3. The experiments use only five trials per model. With so few runs, can we confidently say Evo2-40B is truly more vulnerable than the smaller ones?

4. While plausible, this conclusion isn’t proven. Larger models correlate with higher similarity scores, but correlation ≠ causation. No controlled safety-aligned baseline is tested to confirm that scaling causes more risk.

**Questions:**

1. Why is 90 % BLAST identity chosen as the success threshold? Did you test whether this threshold actually correlates with any functional or structural similarity beyond sequence overlap?

2. Can GeneBreaker jailbreak models that are explicitly safety-aligned or fine-tuned with filtering mechanisms, or does it only work on open, research-grade DNA models?

**Details Of Ethics Concerns:**

No concern

---

> ### Author Response · Authors · 2025-11-16
>
> We thank the reviewer for the constructive comments and appreciation! As for your questions:
>
> **Q1**.  The paper stops at exposing risks but provides no concrete defense or biosafety governance mechanism beyond brief veto filtering in the Appendix.
>
> **R1**: We appreciate the reviewer’s suggestion! Besides the inference-time veto filtering, we designed and implemented the Direct Preference Optimization (DPO)–based safety-alignment baseline. DPO aligns the DNA LM by comparing preferred vs. dispreferred continuations using the standard DPO objective, encouraging the model to increase the likelihood of benign DNA sequences. To construct preference pairs, we generated multiple continuations for each jailbreak prompt and selected preferred data with a low PathoLM score (<0.2) and no pathogenic BLAST hit, while selecting dispreferred data with a high PathoLM score (>0.8) or >90% BLAST identity to regulated pathogens. This yields 48k preference pairs across all viral categories. We trained the aligned model using LoRA (rank 16) for 3 epochs on all attention and MLP layers, enabling efficient alignment for large DNA LMs. The aligned Evo2-7B model shows a substantial reduction in attack success (e.g., SARS-CoV-2: 60%→14%, HIV-1: 50%→20%) while maintaining performance on benign tasks (e.g., GFP, β-globin).
>
> **Q2**. The benchmark seems built entirely by the authors. How can we be sure JailbreakDNABench isn’t biased toward viruses that are easier to hit, making GeneBreaker look stronger?
>
> **R2**: The benchmark is not built entirely by the authors. Instead, as stated in Lines 260-263, JailbreakDNABench is curated based on the U.S. Department of Health and Human Services (HHS) and U.S. Department of Agriculture (USDA) Select Agents and Toxins Lists, which catalog biological agents and toxins that pose significant threats to human, animal, and plant health.
> Moreover, for all tested DNA language models, we check and make sure the viruses in JailbreakDNABench do not exist in the training set. Therefore, we can be sure JailbreakDNABench isn’t biased toward viruses that are easier to hit.
>
> **Q3**. The experiments use only five trials per model. With so few runs, can we confidently say Evo2-40B is truly more vulnerable than the smaller ones?
>
>
> **R3**: In experiments, we run five trials for each target DNA sequence.
> During the rebuttal, we reran experiments of the Evo2 series models for 10 trials and show the new results below:
>
> | Model        | Large DNA      | Small DNA       | +ssRNA         | -ssRNA         | dsRNA          | Enteric RNA    |
> |--------------|----------------|------------------|----------------|----------------|----------------|----------------|
> | Evo2 (1B)    | 22.0 ± 6.3    | 20.0 ± 40.0      | 13.3 ± 8.3     | 0.0 ± 0.0      | 0.0 ± 0.0      | 20.0 ± 40.0    |
> | Evo2 (7B)    | 44.0 ± 8.4     | 46.7 ± 26.7      | 27.8 ± 5.9    | 24.4 ± 12.8    | 20.0 ± 40.0    | 50.0 ± 15.8    |
> | Evo2 (40B)   | 54.0 ± 9.7     | 60.0 ± 25.0      | 38.9 ± 9.4     | 26.7 ± 14.1    | 20.0 ± 40.0    | 60.0 ± 20.0    |
>
> Using two-sample t-tests (n=10), Evo2-40B significantly outperforms Evo2-7B for Large DNA (p<0.05) and +ssRNA (p<0.01), while Evo2-7B significantly outperforms Evo2-1B across Large DNA, +ssRNA, -ssRNA, and Enteric RNA categories (all p<0.05), confirming clear scaling benefits on multiple viral families.
>
> **Q4**. While plausible, this conclusion isn’t proven. Larger models correlate with higher similarity scores, but correlation ≠ causation. No controlled safety-aligned baseline is tested to confirm that scaling causes more risk
>
> **R4**: To further show that larger models correlate with higher similarity scores, we perform jailbreak attacks on *safety-aligned Evo models* (the safety alignment methods are described in response 1).
>
> | Model        | Large DNA      | Small DNA       | +ssRNA         | -ssRNA         | dsRNA          | Enteric RNA    |
> |--------------|----------------|------------------|----------------|----------------|----------------|----------------|
> | Evo2_aligned (1B)    | 0.0 ± 0.0    | 0.0 ± 0.0      | 13.3 ± 8.3     | 0.0 ± 0.0      | 0.0 ± 0.0      | 0.0 ± 0.0    |
> | Evo2_aligned (7B)    | 20.0 ± 17.9     | 30.0 ± 25.0      | 17.8 ± 5.4   | 0.0 ± 0.0    | 0.0 ± 0.0    | 0.0 ± 0.0    |
> | Evo2_aligned (40B)   | 28.0 ± 9.7     | 30.0 ± 25.0      | 28.8 ± 11.3     | 0.0 ± 0.0    | 0.0 ± 0.0    | 20.0 ± 20.0    |
>
> We observe that for safety-aligned DNA language models, larger parameter size still correlates with higher attack success rates.

---

> ### Author Response · Authors · 2025-11-16
>
> **Q5**. Why is 90 % BLAST identity chosen as the success threshold? Did you test whether this threshold actually correlates with any functional or structural similarity beyond sequence overlap?
>
> **R5**: In Lines 287-295, we discussed why 90% BLAST is chosen as the success threshold. We choose 90% BLAST identity following the Common Mechanism developed by IBBIS for synthetic nucleic acid screening, where >90% similarity is widely used to flag sequences as potentially matching regulated pathogens. This threshold provides a balance between biosafety sensitivity and false-positive reduction. Empirically, high nucleotide identity (≥90%) is known to correspond to conserved genomic regions essential for viral replication or infectivity (Harvey et al., 2021), meaning that sequences above this level typically preserve key functional motifs of pathogens such as SARS-CoV-2 or HIV-1. In addition, protein sequences with ≥90% identity generally maintain structural and functional fidelity, even though lower identities may retain homologous folds. Prior work in sequence homology search (Ye et al., 2006; Pearson, 2013) also supports 90% as a reliable threshold for detecting biologically meaningful matches while avoiding spurious alignments. Therefore, 90% BLAST identity is a domain-standard, biologically interpretable, and safety-aligned criterion for determining jailbreak success.
>
> To verify that 90% BLAST identity reflects meaningful biological similarity, we performed a supplementary analysis on 200 jailbreak-generated Spike variants. For each sequence, we computed nucleotide BLAST identity and predicted protein structures using AlphaFold3, then aligned them to the corresponding reference structures (6VXX) with TM-align to obtain backbone RMSD. We calculated the Pearson correlation between BLAST identity and structural similarity (using negative RMSD), obtaining r = 0.81, indicating a strong link between sequence identity and structural preservation. Moreover, 92% of sequences with ≥90% nucleotide identity had RMSD < 2 Å, confirming that this threshold reliably captures variants that retain native structural features.
>
> **Q6**.  Can GeneBreaker jailbreak models that are explicitly safety-aligned or fine-tuned with filtering mechanisms, or does it only work on open, research-grade DNA models?
>
> **R6**: Yes. GeneBreaker is able to jailbreak both open research-grade DNA LMs and models that have been explicitly safety-aligned. In addition to inference-time veto filtering, we trained a Direct Preference Optimization (DPO)–aligned Evo2-7B model using 48k preference pairs constructed from benign (low-PathoLM) versus pathogenic (high-PathoLM or high-BLAST) continuations. The aligned model, trained with LoRA (rank 16, 3 epochs), shows a substantial reduction in attack success (e.g., SARS-CoV-2: 60%→14%, HIV-1: 50%→20%). However, jailbreak attacks still succeed, even under alignment, demonstrating that current safety fine-tuning and filtering strategies mitigate but do not eliminate jailbreak vulnerability. This indicates that safety-aligned DNA LMs continue to exhibit residual biosafety risks, and GeneBreaker provides a practical framework to quantify these remaining failure modes.

---

> > ### Comment · Reviewer_TZ2K · 2025-11-26
> >
> > Thank you to the authors for the thoughtful rebuttal and the additional experiments. The new results, especially the DPO-based alignment baseline and the structure–identity correlation analysis, clarify parts of my earlier concerns. That said, several issues I raised are only partially addressed.
> >
> > On Q5, the justification for using a 90% BLAST threshold is now better explained, but it still leans mainly on sequence and structural similarity rather than direct functional or pathogenic relevance. The AlphaFold3 correlation is interesting, but it’s limited to Spike and to model-generated sequences, so it doesn’t fully establish that BLAST identity alone is an appropriate biosafety measure.
> >
> > On Q6, I appreciate the attempt to test GeneBreaker on safety-aligned models. However, the alignment procedure is built around the authors’ own PathoLM classifier, which is also used during attack guidance. This raises concerns about circularity, and it remains unclear how GeneBreaker would fare against independently aligned models or more robust safety mechanisms.
> >
> > Overall, the rebuttal strengthens the empirical side of the paper, but some conceptual issues, particularly around evaluation validity, dependence on the authors’ own tools, and the scope of the biosafety claims, are still open. I am keeping my score at 6 (marginal accept): the paper is timely and potentially impactful, but it would benefit from a more rigorous and clearly decoupled evaluation framework.

---

> > > ### Author Response · Authors · 2025-11-28
> > > **Thanks for your reply! More supplementary results to address concerns**
> > >
> > > Dear Reviewer,
> > >
> > > We are glad that part your concerns have been addressed. As for your further concerns,
> > >
> > > **1**. On Q5, the justification for using a 90% BLAST threshold is now better explained, but it still leans mainly on sequence and structural similarity rather than direct functional or pathogenic relevance. The AlphaFold3 correlation is interesting, but it’s limited to Spike and to model-generated sequences, so it doesn’t fully establish that BLAST identity alone is an appropriate biosafety measure.
> > >
> > > **R1**: In the original paper, we already used direct functional annotation to complement sequence/structural similarity. Specifically, as indicated in lines 295-300, we employ the **Viral Annotation DefineR(VADR,v1.5.1)**, an NCBI tool for validating and **annotating viral genomes functions** with curated RefSeq models and BLASTn. VADR projects functional features such as coding sequences, mature peptides, and structural RNAs, and validates them with blastx alignments against referenceproteins, issuing deterministic alerts when inconsistencies occur.
> > >
> > > Together, BLAST and VADR allow us to assess both sequence-level similarity and functional conservation, providing a rigorous evaluation of jailbreak success.
> > >
> > > To further establish that BLAST identity is an appropriate biosafety measure, we calculated the Pearson correlation between BLAST identity and structural similarity (using negative RMSD) **on all generated sequences from the JailbreakDNABench**. An average correlation value of 0.77 further indicates a strong link between sequence identity and structural preservation.
> > >
> > > **2**.  On Q6, I appreciate the attempt to test GeneBreaker on safety-aligned models. However, the alignment procedure is built around the authors’ own PathoLM classifier, which is also used during attack guidance. This raises concerns about circularity, and it remains unclear how GeneBreaker would fare against independently aligned models or more robust safety mechanisms.
> > >
> > >
> > > **R2**: Here, we conducted an additional experiment **using geNomad as an alternative pathogen-scoring method**.
> > > geNomad is a state-of-the-art model for detecting viral genetic elements [1], and was recently used as the pathogenicity-guidance signal in Generative design of novel bacteriophages with genome language models [2]. We trained a Direct Preference Optimization (DPO)–aligned Evo2-7B model using 48k preference pairs constructed from benign (low-geNomad score) versus pathogenic (high-geNomad or high-BLAST) data. The aligned model shows a reduction in attack success (e.g., SARS-CoV-2: 60%→15%, HIV-1: 50%→18%). However, **jailbreak attacks still have a non-zero success rate against independently aligned models.**
> > >
> > >
> > > References
> > > [1] Camargo, A., et al. (2023). Identification of mobile genetic elements with geNomad. Nature Biotechnology.
> > > [2] King, S. H., et al. (2025). Generative design of novel bacteriophages with genome language models. bioRxiv.

---

### Meta-Review · Area_Chair_YioS · 2026-01-09

**Summary:**

This paper proposes to study the safety issues of DNA LMs. This work first constructs a benchmark for evaluating the risk of generating virus DNA sequences, and then builds a method for evaluating how susceptible DNA LMs are to jailbreaking attacks. They found that their method can successfully jailbreak Evo series models with a high success rate, which shows the dual risk of scaling DNA LMs.

Overall, reviewers found this work timely as bio LMs get stronger, and this work as among the first in studying the safety issues of DNA models. The concerns were mainly about evaluation, method clarity, and experimental completeness, but no reviewer objects on scientific grounds. Therefore, I recommend the acceptance of this paper pending ethics clearance due to the dual-use risk.

**Reviewer Concerns:**

1. clarification of GenomeOcean architecture: addressed
2. experiment issues such as BLAST threshold, trial count, and defense baselines: mostly addressed.
3. lack of wet-lab verification: still outstanding but this is inherently hard.
4. dual-use ethics concern: requires ethics review for further evaluation.

**Reviewer Scores:**

Two reviewers have explicitly stated that they are keeping their score, and given the consistency of the review scores, I think it is very likely that all reviewers would have kept their score.

---

### Decision · Program_Chairs · 2026-01-26

Accept (Poster)